# A Narrative Literature Review of Bias in Collecting Patient Reported Outcomes Measures (PROMs)

**DOI:** 10.3390/ijerph182312445

**Published:** 2021-11-26

**Authors:** Michela Luciana Luisa Zini, Giuseppe Banfi

**Affiliations:** 1School of Medicine, Università Vita-Salute San Raffaele, 20132 Milan, Italy; banfi.giuseppe@hsr.it; 2IRCCS Istituto Ortopedico Galeazzi, 20161 Milan, Italy

**Keywords:** PROMs, PROs, patient reported outcomes, patient reported outcomes measures, bias

## Abstract

There is a growing interest in the collection and use of patient reported outcomes because they not only provide clinicians with crucial information, but can also be used for economic evaluation and enable public health decisions. During the collection phase of PROMs, there are several factors that can potentially bias the analysis of PROM data. It is crucial that the collected data are reliable and comparable. The aim of this paper was to analyze the type of bias that have already been taken into consideration in the literature. A literature review was conducted by the authors searching on PubMed database, after the selection process, 24 studies were included in this review, mostly regarding orthopedics. Seven types of bias were identified: Non-response bias, collection method related bias, fatigue bias, timing bias, language bias, proxy response bias, and recall bias. Regarding fatigue bias and timing bias, only one study was found; for non-response bias, collection mode related bias, and recall bias, no agreement was found between studies. For these reasons, further research on this subject is needed in order to assess each bias type in relation to each medical specialty, and therefore find correction methods for reliable and comparable data for analysis.

## 1. Introduction

Patient Reported Outcomes can be defined as “any report of the status of a patient’s health condition that comes directly from the patient, without interpretation of the patient’s response by a clinician or anyone else. The outcome can be measured in absolute terms (e.g., severity of a symptom, sign, or state of a disease) or as a change from a previous measure” [1].

There is a growing interest in the collection and use of PROs, which means that medicine is increasingly focusing on the patients’ perspective and needs. Patient reported outcomes were initially developed and used in research [2,3], but today, they are serving wider purposes: They provide clinicians with crucial information, supporting them in decision making [1]. They can also be used to compare different treatments and for economic evaluation: PROM results enable public health decisions assessing the cost-effectiveness and cost utility from the patient perspective [4,5].

Patient reported outcomes can be collected by different methods: In person during clinical visits using a paper questionnaire, via telephone, online using a digital questionnaire; they can also be self-administrated or assisted.

There are several factors during the collection phase of PROMs that can potentially bias the analysis of PROM data. It is crucial that the collected data are reliable and comparable. In this paper, we wanted to analyze the types of bias that have already been considered in the literature.

The primary aim of this review was to identify potential bias in data collection that can influence the reliability and objectivity of PROM data analysis. Bias is intended as a possible external factor that can modify the accuracy of the PROM data collection and interpretation. The biases described in the present review are defined by the authors of different papers and are not exhaustive, some additional factors could be discovered and evaluated on the current and future PROMs including proxy measures possibly covered by PROMs.

## 2. Materials and Methods

During May 2021, a literature review was conducted by the authors searching “PROMs” and “Patient reported outcomes AND bias” on the PubMed database considering the years 1990 to 2021, in the English language. No exclusion criteria were used.

The selection process is shown in Figure 1, the search led to the identification of 3295 studies, 3261 after removing duplicates. The first step was to conduct a title screening, where 3193 titles were excluded because they were not consistent with the objective of the analysis. The 68 abstracts thus selected were read and analyzed, and only 37 of them were assessed for eligibility. After the full-text reading, 13 studies were excluded because seven of them were not consistent with the objective and six of them dealt with the subject only in a marginal way. A final number of 24 studies were included in the review.

## 3. Results

The search led to the analysis of 24 studies, as shown in Table 1.

It is interesting to note that most of these are related to orthopedic patients (Figure 2).

The identified articles are fairly recent, published from 2004 to 2021 (Figure 3), most of them in the last five years, suggesting the subject is relatively new, but that there is a growing interest in it.

Most of the studies on bias in collecting PROMs were carried out in the USA (10), followed by five studies led in the UK (Figure 4).

The review identified seven types of bias (Figure 5): Collection method related bias, non-response bias, proxy response bias, recall bias, language bias, fatigue bias, and timing bias.

### 3.1. Collection Mode Related Bias

PROMs may be collected in different ways including in-person surveys, phone calls, online surveys, and paper surveys. The specific method of PROM collection may bias the results, especially acquiescence (or condescending) bias. In fact, patients may provide responses according to what they believe the researchers would like to collect rather than the most accurate and true answers [6].

Nine results were found (Table 2), where two of them were meta-analyses; most of them were related to the orthopedic field.

Four papers agreed on the conclusion that scores reported on the phone with an interviewer were significantly better than the ones self-administrated and collected online, in person or paper based, suggesting the presence of a human interviewer can possibly bias the results. This is in contrast to the analysis of Rutherford et al., who also assessed self-completed versus assisted PROM collection and found no significant differences between the two methods.

One study compared the traditional paper based collection method with the use of an app for smartphones: No difference between the two methods was shown, neither at baseline nor at the follow-up visits, demonstrating that it is feasible to use an app as a collection method as a valid alternative to a paper based one [17].

Four papers, one of which was a meta-analysis, considered the paper based collection method versus the online collection method: Overall, all of them did not find any significant difference between the two methods, but one found differences in scores on the EQ-5D descriptive questions, pain visual analog scale (VAS), and the NDI [20].

In conclusion, we can see how these papers disagree on whether or not the presence of a human interviewer can possibly bias the PROM results; on the other hand, no difference was found between data collection with the use of an app or online and the paper based questionnaire.

### 3.2. Non-Response Bias

Non-response bias, which can sometimes also be addressed as “selection bias”, is related to patients that quit and do not fill in the post-operative PRO questionnaires.

Non-response bias is difficult to assess because, by definition, the data of non-respondents are not in the records, so non-response to the PRO questionnaire can introduce a bias if the respondents are not representative of non-respondents [7], and therefore it is important to understand whether there are reasons why these patients do not follow up and what their perception would be.

This bias was taken into consideration and studied in eight different papers (Table 3), all related to the orthopedic field, one of them also considering varicose vein surgery.

Different methods have been used to assess non-response bias.

Two papers considered late respondent patients as proxies of non-responders. Late responders were identified as patients who did not respond to the first request to fill in the questionnaire, but only responded to the last reminder [7] or as patients who responded after a certain time post-surgery (4.5 months for hernia repair and VV surgery and 7.5 months for hip and knee replacements) [23].

Both these studies concluded that late responders tend to report poorer outcomes, and this can lead to overestimating the hospital performance.

Five papers identified non-responders as patients who failed to complete the questionnaire and were later contacted by telephone or email. Only Kim et al. concluded that non responders reported worst outcomes than responders as well as that there was a significant association between patient satisfaction and the rate of response: Early responders were more frequently very satisfied, while non-responders reported they “wished it was better.” [22]. On the other hand, one paper reported that non-responders introduced a bias, but in contrast, patients who stopped follow-up reported better scores [13]. Three papers did not find any statistically significant difference between responders and non-responders, suggesting that people leaving the follow-up would not bias the results in either way, for better or worse, even if one of these reported a better patient satisfaction among responders [14]. One paper also identified a trend of worst outcomes for non-responders [12].

Finally, one paper did not contact non-responders nor used late responders as proxies. They identified four postoperative time points at six-week, six-months, 12-months, and 24-months. At each time point, patients were classified as responders and non-responders. At every time point, the authors analyzed what the non-responders score was at the previous time point when compared to the responders. The study showed how non-response to the follow-up surveys does not appear to be random, those who did not respond had significantly lower scores than responders, and so patients reporting good outcomes were overrepresented [18].

In relation to the socio-demographic factors, five papers agreed that younger patients were more likely to stop follow-up or be late responders [12,13,14,18,23]. Moreover Hutchings et al. found that late responders were more often non-white and socially deprived. [23].

We can see how authors have disagreed on whether or not non-responders can bias PROM analysis. It is important to further study this bias, because the risk to assess is failing to identify poor performing hospitals, in fact, if good outcomes are overrepresented, this can lead to overestimating the hospital performance.

### 3.3. Proxy/Caregiver Response Bias

Proxies and caregivers are people who answer questions on the patient’s behalf when the patient is incapable of responding themselves (for example, due to physical or cognitive problems or among elderly or disabled patients) [30]. Caregiver/proxy bias can be introduced when the caregiver completes patient-reported outcome questionnaires because they might rate symptoms better or worse than the patients themselves. [31]

Three articles were found in this review regarding proxy response bias (Table 4), all of them from the USA.

Lapin et al. and Li et al. agreed that proxy response bias varied depending on the specific domain being tested.

Lapin et al. found higher agreement regarding physical function, pain interference, and sleep disturbance; on the other hand, lower agreement was found in cognitive function, anxiety, and depression. Even if, on average, caregivers reported worse outcomes compared to patient self-report, discrepancies were also found in the other direction with 16–24% of proxies reporting better outcomes than the patients.

A similar conclusion was reported by Li et al., in fact, proxy response bias was present in the physical, affective, cognitive, and social status domains, but not in the sensory status domain. Specifically, proxies tended to report more health and functional limitations [27].

Alvarez et al. found that there was a good agreement in both PROMIS PF and PROMIS PI, even if for PROMIS PI, the proxies reported a slightly higher score. They also discovered that the correlation of PF between the patient and the proxy for younger and older patients was similar; moreover, patients who had daily contact with their proxies and patients who had less contact with their proxies had a similar degree of correlation for PROMIS PF.

In conclusion, we can say that it is harder for proxies to provide accurate response in relation to less observable domains because they might not have all the information regarding patient feelings and private life. On the other hand, it is easier to report objective, observable facts such as difficulties in walking. When developing a survey, it must also be suitable for proxies, and so objective questions are preferred.

Moreover, caregivers generally tend to report worse outcomes, but there are some cases in which the opposite is true and the proxies report a better outcome than the patients, so even if at a group-level the difference is small, when using proxy responses at individual level, caution should be taken.

### 3.4. Recall Bias

Recall bias is related to the possibility that the patients would not exactly recall their preoperative symptoms months after surgery, and so that PROMs collected relying only on patient memory may be better or worse than the one including a preoperative baseline.

Recall bias arises for different reasons, for example, some details may go unnoticed and never be stored in the patient’s memory, or in contrast, the patient might have some new information and with it distort the memory [32].

Three articles were found in this review regarding recall bias (Table 5). Two papers from Aleem et al. had the same goal: To assess the accuracy of patient recollection of preoperative symptoms. One paper focused on cervical spine surgery, the other on lumbar decompression and fusion. Both studies used the same method: Comparing the patients’ responses to a questionnaire taken one year after surgery about their preoperative condition with actual preoperative baseline scores. In both studies, the data indicated that patient recollection of preoperative status was more severe than their actual preoperative status, so the authors suggested that data based on the patients’ recollection are likely to be biased, and it is not advisable to rely on patient recollection of the preoperative state [24,28].

A different conclusion was drawn by Kwong et al., who reviewed the evidence for using retrospective patient-reported outcome measures to assess recall bias in emergency admissions to hospitals. They found a strong association between retrospective and contemporary PROMs and concluded that retrospective collection could be a new way of assessing PROMs in emergency admissions, where it is hard to collect preadmission patient-reported outcome measures due to the unpredictability of the occurrence of illnesses [29].

### 3.5. Language Bias

Questions can be ambiguous for patients as the same question can be interpreted differently by different patients, which could compromise the results.

This type of bias was analyzed in two papers, related to the orthopedic field, both in the UK.

One analyzed the semantic of “hip pain” in patient-reported outcome measures and how this could have different interpretations for patients and clinicians and therefore introduce a bias. Patients often claim to have hip pain, however, by saying “hip”, they are referring to variable structures located in the hip region that are not necessarily related to the hip joint itself.

In this study, seven different anatomical sites were described: Trochanter, hip joint, iliac crests, lumbar spine, sacroiliac joint, and in only 20.8% of cases, ‘hip pain’ had an anatomical relationship to the hip joint itself.

This suggests a clear ambiguity of the semantics of the term ‘hip pain’, and therefore when using these terms in PROMs, this may introduce a bias as patients may be scoring pain that is not related to the hip joint [10].

The other paper focused on question 5 of the Oxford Hip Score (OHS), and how this can be misinterpreted. They discovered that question 5 of the Oxford Hip Score was ambiguous to 10% of native English-speakers, thus causing an error of 8% on the OHS, which may lead to an underestimation of the patient’s hip score [11].

Both papers agreed that PROM questionnaires can be ambiguous for patients and therefore bias the results, so it would be advisable to formulate them in the most accurate way.

### 3.6. Timing Bias

PROMs can be used not only for clinical evaluation, but also to make economic evaluations. They can be collected at different times for postoperative follow-up, but using a different timing could bias the economic evaluation [9].

This type of bias has only been analyzed in one paper based in Australia by Schilling et al. (Table 6) for total knee arthroplasty (TKA).

The authors aimed to study whether the timing of the HRQOL (health-related quality-of-life) measurement could lead to a bias in QALY (quality-adjusted life-years) estimation.

The authors found that postoperative PROM collection at six weeks biased the estimation of QALY gain by 41%, at six months by 18%, and at 12 months by 8%. This bias was minimized (6% error) when collecting PROMs at three months after surgery.

For total knee arthroplasty, they recommend two post-surgery measures: One in the first six months, and the other at 12 months, with linear interpolation between them. If only a single postoperative measurement is possible, they recommend measuring it at three months.

It should also be considered that the optimal timing for economic evaluation may not be the same as for the clinical evaluation [9].

### 3.7. Fatigue Bias

The fatigue bias is related to the length of the collection process; this occurs when the patient, at some point during the survey, becomes tired and starts to give inaccurate answers [7].

It is important to understand whether the quality of answers decreases over time, compromising data reliability.

Only one paper from Cabitza et al. studied fatigue bias at the IRCCS Orthopedic Institute Galeazzi in Milan, Italy.

The study pointed out that less than 20% of patients found the interview to be demanding, among those, only 22% found the interview to be too long, and the remaining 78% found it demanding, but acceptable in length.

The authors also compared the SF Mental, SF Physical score, and VAS pain score of patients who found the interview demanding and of the ones who did not find the interview to be demanding, they found that patients who did not report fatigue scored better than the patients who reported fatigue.

The study shows how worse outcomes were reported from patients who found the PRO interview demanding: This may occur because patients who are in a worse health condition may perceive an excessive length of the interview.

The authors concluded that fatigue bias can affect patient reported outcomes, although slightly. Therefore, the shorter the PRO interview, the higher the data reliability.

## 4. Discussion

What was found in this review is that, first of all, further studies are needed as there was no agreement among the researchers for the three bias types analyzed, and for two bias types, only one result was found.

Anyway, when developing PROM questionnaires, it is recommended that:They are formulated in the most accurate way, in order to avoid language bias;They are not too long, to minimize fatigue bias;The questions are objective and referred to as observable facts, so that caregivers can also answer them in a reliable way.

Our review had some limitations. One limitation to this review is that, in most of the studies analyzed, socio-demographic factors were discussed only marginally, while some factors such as age, education level, and economic conditions could potentially introduce some bias. Furthermore, health-related factors such as neurological disease might influence the patient perception of the disease and therefore their response. We remark that the search was addressed toward PROMs used for physical conditions by using the PubMed database, thus psychosocial conditions were not included.

Another limitation to this review is that it does not meet the criteria of a systematic review. Moreover, the types of bias were identified from the literature; it is possible that other types of bias were missing from the included studies and could be found in the broader PROM literature.

Therefore, all these factors should be investigated in future studies and further research could be conducted.

## 5. Conclusions

Considering the growing importance of PROMs in medicine and public health, we should outline that the global number of experimental papers devoted to the quality and reliability of PROM collection, identification, elaboration, and interpretation are, in general, quite few.

It is crucial to increase the studies in the field to homogenize the collection of PROMs, which is mandatory to improve the comparison among different experiences and to plan new implementations. The homogeneous data could be used to compare different teams, hospitals, health care systems, and, finally, to calculate QALYs, as demonstrated, for example, by the PaRIS program conducted by the OECD for hip and knee replacement surgery [5].

Moreover, the studies concerning different characteristics of PROMs are important to discriminate between clinically relevant outcomes and non-relevant ones, especially for the items correlated with the interrelationship between patients and health care personnel and organization and with the recruitment of patients over time, which is a common problem and criticism for the value based health care programs based on PROMs.

Further investigation of issues related to bias are acknowledged to improve the implementation and interpretation of PROMs. Furthermore, the identified bias, described in the literature, should be considered by proponents of new PROMs, and properly evaluated by scientific associations and regulatory agencies when applied in drug and medical device clinical trials.

Our review, the first in the field, hopes to encourage the scientific and social debate on the argument and improve the use of PROMs in medicine.

## Figures and Tables

**Figure 1 ijerph-18-12445-f001:**
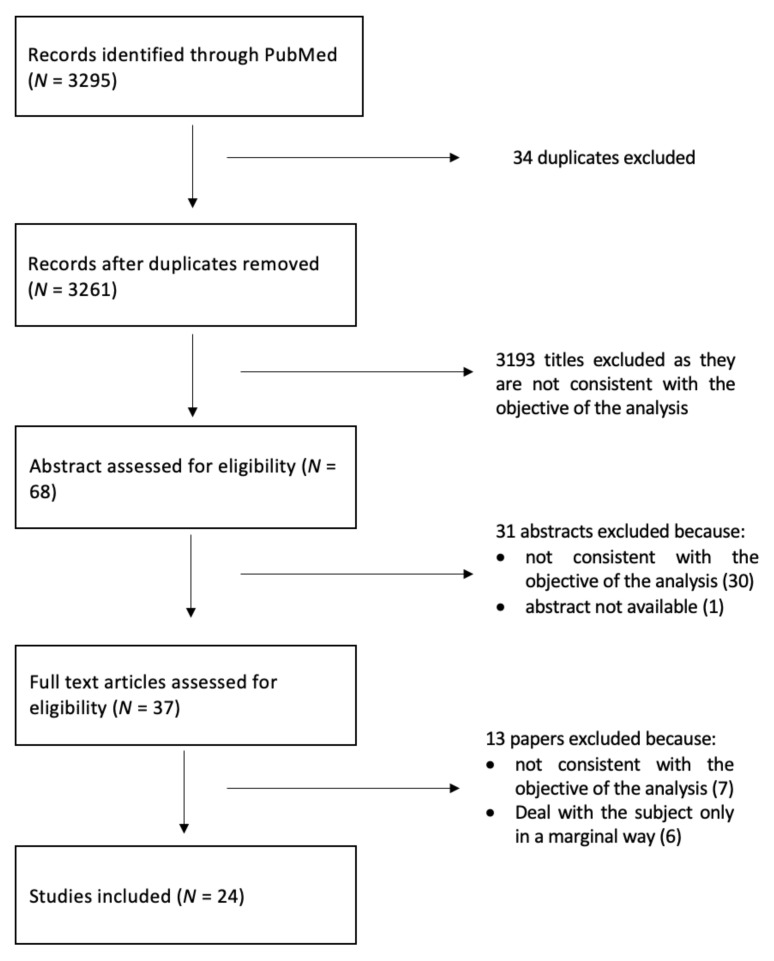
Flow diagram for the search and selection process.

**Figure 2 ijerph-18-12445-f002:**
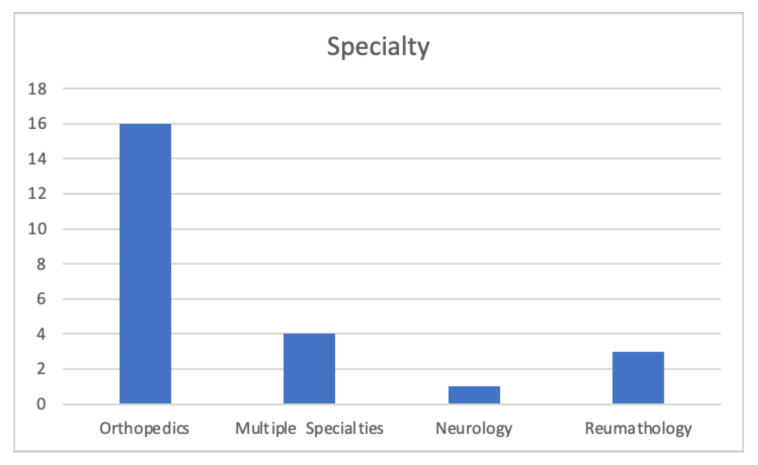
Specialty distribution.

**Figure 3 ijerph-18-12445-f003:**
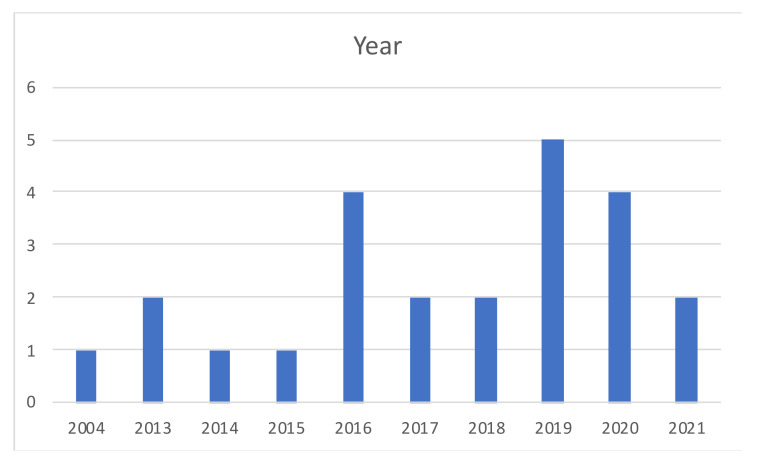
Year distribution.

**Figure 4 ijerph-18-12445-f004:**
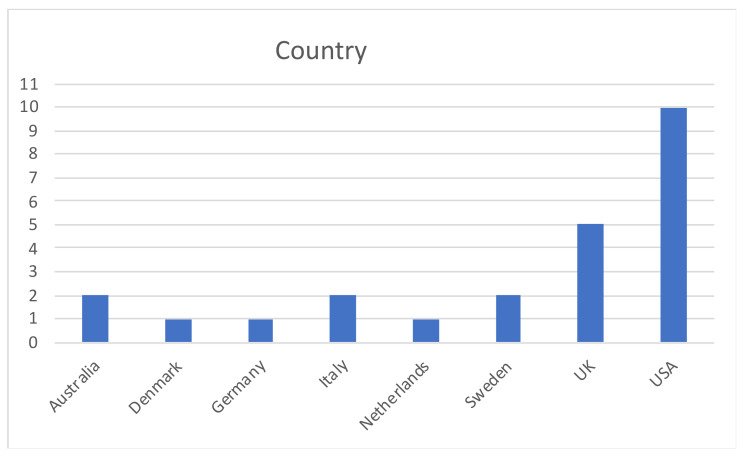
Country distribution.

**Figure 5 ijerph-18-12445-f005:**
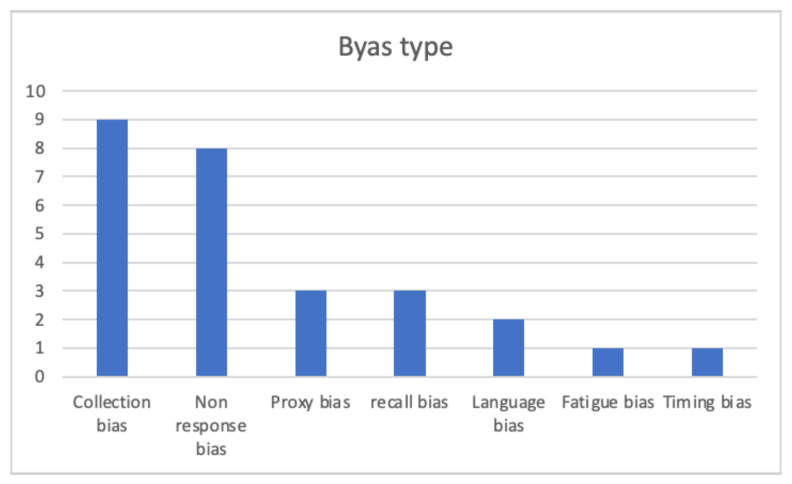
Bias type distribution.

**Table 1 ijerph-18-12445-t001:** Results.

First Author	Year	Country	Bias	Specialty
Cabitza [6]	2018	Italy	Non-response bias, collection bias	Orthopedics
Cabitza [7]	2019	Italy	Non-response bias, collection bias, fatigue bias	Orthopedics
Hammarstedt [8]	2017	USA	Collection bias	Orthopedics
Schilling [9]	2016	Australia	Timing bias	Orthopedics
Karela [10]	2020	UK	Language bias	Rheumatology
Péchon [11]	2019	UK	Language bias	Orthopedics
Polk [12]	2013	Denmark	Non response bias	Orthopedics
Chen [13]	2020	USA	Non response bias	Orthopedics
Lindman [14]	2020	Sweden	Non response bias	Orthopedics
Acosta [15]	2020	USA	Collection bias	Orthopedics
Lapin BR [16]	2021	USA	Proxy bias	Neurology
Richter [17]	2021	Germany	Collection bias	Rheumatology
Imam [18]	2014	UK	Non response bias	Orthopedics
Schröder [19]	2019	Netherlands	Collection bias	Orthopedics
Shah [20]	2016	USA	Collection bias	Orthopedics
Rutherford [21]	2016	Australia	Collection bias	Multiple specialties
Kim [22]	2004	USA	Non response bias	Orthopedics
Hutchings [23]	2013	UK	Non response bias	Multiple specialties
Aleem [24]	2017	USA	Recall bias	Orthopedics
Hofstedt [25]	2019	Sweden	Collection bias	Rheumatology
Alvarez-Nebreda [26]	2019	USA	Proxy bias	Orthopedics
Li [27]	2015	USA	Proxy bias	Multiple specialties
Aleem [28]	2018	USA	Recall bias	Orthopedics
Kwong [29]	2016	UK	Recall bias	Multiple specialties

**Table 2 ijerph-18-12445-t002:** Collection mode related bias results.

1st Author	Collection Methods	Results
Cabitza [6]	Electronic/Online (self-administrated) Telephone (with interviewer)	Collection methods can bias PROM scores: Scores reported on the phone are better than the ones reported online.
Cabitza [7]	Electronic/Online (self-administrated) Telephone (with interviewer)	Collection methods can bias PROM scores: Scores collected on the telephone with an interviewer are significantly better than those collected online.
Hammarstedt [8]	In person (self-administrated) Electronic/Online (self-administrated) Telephone (with interviewer)	Collection methods can bias PROM scores: PROMs collected via telephone are higher than online and in person.
Acosta [15]	Telephone Electronic/Online Paper based In person	Collection method can bias PROM scores: PROMs collected via telephone are higher than online and paper.
Richter [17]	App Paper based	No significant differences between paper based and online method.
Schröder [19]	Paper based Electronic/Online	No significant differences between paper based and online method.
Shah [20]	Paper based Electronic/Online	No significant differences between paper based and online method. Differences in scores on the EQ-5D descriptive questions, pain visual analog scale (VAS), and the NDI.
Rutherford [21]	Paper based vs. Electronic/Online Self-completed vs. Assisted	No significant differences between self-completed paper and electronic. Self-completion and assisted completion generate equivalent scores overall.
Hofstedt [25]	Paper based Online	No significant difference between paper based and online method.

**Table 3 ijerph-18-12445-t003:** Non-response bias results.

First Author	PROMs and Questionnaire Used	Surgery Type	Results
Cabitza [6]	SF-12 Mental and Physical SF-36 Mental and Physical	Mostly hip and knee prosthetic surgery and spine- related procedures.	There is no evidence that people quitting the follow-up (PRO) program would create either significantly better or worse scores.
Cabitza [7]	SF-12 Mental and Physical SF-36 Mental and Physical VAS pain score	Mostly hip and knee prosthetic surgery and spine- related procedures.	Early responders reported a lower pain and better outcomes than late responders. But no difference for SF Mental Score.
Polk [12]	Western Ontario Osteoarthritis of the Shoulder (WOOS) index	Shoulder replacement	Non-responders did not bias the overall results, but there is a trend of worst outcome for non-responders.
Kim [22]	Modification of the Knee Society clinical rating system and functional score.	Total Knee Arthroplasty	Non-responders introduce a bias: they report poorer outcomes than responders.
Hutchings [23]	Oxford Hip Score Oxford Knee Score Aberdeen Varicose Vein Questionnaire EQ-5D (EuroQuol 5 Dimensions) Sociodemographic questionnaires	Varicose vein surgery Hernia repair Knee or hip replacement	Non-responders introduce a bias: late responders reported a slightly poorer outcome (not statistically significant for VV surgery).
Chen [13]	ODI (Oswestry Disability Index) VAS (Visual Analog Scale) leg/back Short Form-12 (SF-12) Physical/Mental PROMIS	Spine surgeries (decompression or fusion)	Non-responders introduce a bias: patients who stop follow-up score better.
Lindman [14]	HAGOS (Copenhagen Hip and Groin Outcome Score) EQ-5D (EuroQuol 5 Dimensions) iHOT-12 (International Hip Outcome Tool) HSAS (Hip Sports Activity Scale) VAS (Visual Analogue Scale) for hip function	Hip arthroscopies	Non-responders did not bias the overall results: there is no difference except for patient satisfaction.
Imam [18]	EQ-5D (EuroQuol 5 Dimensions) Oxford Hip Score complication, service satisfaction and outcome satisfaction questionnaires	Total hip replacement	Non responders introduce a bias: patients reporting good outcomes are overrepresented.

**Table 4 ijerph-18-12445-t004:** Proxy/caregiver response bias results.

1st Author	PROMs and Questionnaire Used	Results
Lapin [16]	PROMIS Global Health PROMIS 8-item SF: physical function, satisfaction with social roles and activities, anxiety, fatigue, pain interference, sleep disturbance Patient Health Questionnaire-9 depression screen Neuro-QoL cognitive function	Proxy response bias depends on the domain being tested: it is greater on more subjective domains, with proxies reporting worse outcomes.
Alvarez-Nebreda [26]	PROMIS physical function (PF) and pain interference (PI) Timed Up and Go (TUG) FRAIL Questionnaire	Good agreement in both PROMIS PF and PROMIS PI even if for PROMIS PI proxies report a slightly higher score.
Li [27]	MCBS (Medicare Current Beneficiary Survey)	Proxy response bias depends on the domain being tested: it was present in the physical, affective, cognitive, and social status domains but not in the sensory status domain.

**Table 5 ijerph-18-12445-t005:** Recall bias results.

1st Author	PROMs and Questionnaire Used	Results
Aleem [24]	Numeric Pain Scores (NPS) for back pain and leg pain Oswestry Disability Indices (ODI)	Relying on patient recollection does not provide an accurate measure of preoperative state.
Aleem [28]	Numeric Pain Scores (NPS) for neck pain, arm pain Neck Disability Indices (NDI)	Relying on patient recollection does not provide an accurate measure of preoperative state.
Kwong [29]	SF-36/SF-12; WOMAC; American Urological Association Symptom Index, Western Ontario Meniscal Evaluation Tool; Knee Injury and Osteoarthritis Outcome Score; Oxford Hip Score; Lower Extremity Functional Scale and the feeling thermometer.	Retrospective collection offers a means of assessing PROMs in emergency admissions.

**Table 6 ijerph-18-12445-t006:** Timing bias results.

First Author	PROMs and Questionnaire Used	Surgery Type	Results
Shilling [9]	SF-12	Total knee arthroplasty (TKA)	Timing of PROM collection and the interpolation assumptions can bias economic evaluation.

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
