# Peer review of "A Narrative Literature Review of Bias in Collecting Patient Reported Outcomes Measures (PROMs)"

_ijerph, 2021, doi:10.3390/ijerph182312445_

Round 1
Reviewer 1 Report
Zini and Banfi report an extremely interesting review on biases that may affect the reliability of Patient Reported Outcome Measures.
From their literature search they identify 7 classes of potential bias and briefly discuss their impact on reliability.
Of course, the nature of the biases reviewed vary for their importance depending on the pathology under consideration
The paper could be accepted almost as it stands yet it may benefit of some reorganization
It is not clear at the moment on what basis the order of discussion of the different biases was determined. Yet it is not difficult to see that some biases were more due to organizational /design factors (e.g. collection mode bias; timing bias) while others are related to the properties of the patient samples under investigation and their ability to provide reliable information (recollection bias), or the interaction of these factors thereof.
Perhaps the authors may try to re-organize the review according to one such criteria or more trivially according to the number of available reports (this may also apply to the bar graph in figure 5.
Also a table that indicates how bias is measured for each class might be useful.
In the discussion the Authors may dig a bit deeper into some demographic factors (e.g. age; gender, education) as much as cognitive factors that may impact in generating some biases: for example, it is well known that neurological patients suffer of deficits of awareness of their disorder: as a consequence their RROMs may be less reliable.
In the same vein, the authors may consider and discuss whether a very-short cognitive assessment might help in increasing the reliability of Patient Reported Outcome Measures, at least in prospectively designed clinical experiments, if not in the daily use of PROMs.
A short list of the most urgent recommendations might be an useful completion of this report.
Reviewer 2 Report
The authors report on a systematic review examining bias in the collection of patient-reported outcome measures which is a very interesting and novel topic. However, the certainty of the synthesis and reproducibility of the findings is low as there are few details regarding the methodology and no reference to a published protocol. For example, there is a lack of reporting of the full search strategy for Pubmed, use of a single database, no inclusion criteria, no mention of a certainty assessment, and no reporting of screening and extraction in duplicate. The authors may find the methodology section of the PRISMA checklist useful in reviewing the areas of the methodology that should be considered. I have not fully reviewed other sections of the manuscript as major changes to the methodology are required which may change the reporting and findings in other sections.
Round 2
Reviewer 2 Report
Thank you for changing the title to a narrative review which aligns well with what the authors describe.
I have a some minor points but otherwise support publication of this paper which will pave the way for greater investigation of the types of bias that are highlighted. On this point I wondered whether a little more could be added to the discussion regarding the next steps for further investigation of issues related to bias. Should there be emphasis on this in the development of new PROMs and in the research reporting?
Other points are as follows:
(1) What is meant by "deal with in a minor way" as this could be quite a subjective way of choosing papers that were suitable for inclusion?
(2) Could the authors please define key concepts used in the search strategy including patient-reported outcomes, patient-reported outcome measure and bias as these definitions speak to literature was and was not included? For example, did patient-reported outcome measures cover proxy measures?
(3) The authors should acknowledge that their search was likely biased toward PROMs used for physical conditions rather than psychosocial conditions due to the exclusive use of the Pubmed database.
(4) Could the authors please add further details regarding the process used to identify relevant literature (which aligns with the checklist completed) - how many people were involved in screening and data extraction, years considered, languages included, publication status, and the search strategy? These items speak to how replicable your search and findings might be.
(5) Discussion: Line 615: Not being too long does not guarantee data reliability - could the authors please revise this statement.
(6) Can the authors add some detail regarding how they came up with the types of bias that were included? Were these types of bias identified from literature of did the authors come up with these types of bias from the studies that were identified for inclusion? Were there any other types of bias in broader PROM literature that were missing from the included studies? For example, I noticed confirmation bias was not mentioned which is the tendency for clinicians to favour information that confirms prior beliefs and to discount information that challenges their belief. Also it is well documented that PROM scores may be affected by mood, expectations and prior experience - are these captured by the types of bias mentioned?
